# A systematic review of the clinical profile of patients with bubonic plague and the outcome measures used in research settings

**Josephine Bourner** [1]*, **Lovarivelo Andriamarohasina**[2], **Alex Salam**[1], **Nzelle Delphine Kayem**[1], **Rindra Randremanana**[2], **Piero Olliaro**[1]

**1** ISARIC, Pandemic Sciences Institute, University of Oxford, Oxford, United Kingdom, **2** Institut Pasteur de Madagascar, Antananarivo, Madagascar

* josephine.bourner@ndm.ox.ac.uk

**Data Availability Statement:** All relevant data are within the paper and its Supporting Information files.

## Abstract

### Background

Plague is a zoonotic disease that, despite affecting humans for more than 5000 years, has historically been the subject of limited drug development activity. Drugs that are currently recommended in treatment guidelines have been approved based on animal studies alone–no pivotal clinical trials in humans have yet been completed. As a result of the sparse clinical research attention received, there are a number of methodological challenges that need to be addressed in order to facilitate the collection of clinical trial data that can meaningfully inform clinicians and policy-makers. One such challenge is the identification of clinically-relevant endpoints, which are informed by understanding the clinical characterisation of the disease–how it presents and evolves over time, and important patient outcomes, and how these can be modified by treatment.

### Methodology/Principal findings

This systematic review aims to summarise the clinical profile of 1343 patients with bubonic plague described in 87 publications, identified by searching bibliographic databases for studies that meet pre-defined eligibility criteria. The majority of studies were individual case reports. A diverse group of signs and symptoms were reported at baseline and post-baseline timepoints–the most common of which was presence of a bubo, for which limited descriptive and longitudinal information was available. Death occurred in 15% of patients; although this varied from an average 10% in high-income countries to an average 17% in low- and middle-income countries. The median time to death was 1 day, ranging from 0 to 16 days.

### Conclusions/Significance

This systematic review elucidates the restrictions that limited disease characterisation places on clinical trials for infectious diseases such as plague, which not only impacts the definition of trial endpoints but has the knock-on effect of challenging the interpretation of a

**Funding:** This work was supported by the UK Foreign, Commonwealth and Development Office (https://www.gov.uk/government/organisations/foreign-commonwealth-development-office) and Wellcome (https://wellcome.org/) (Grant ref: 215091/Z/18/Z). The funders had no role in study design, data collection and analysis, decision to publish, or preparation of the manuscript.

**Competing interests:** The authors have declared that no competing interests exist.

trial's results. For this reason and despite interventional trials for plague having taken place, questions around optimal treatment for plague persist.

## Author summary

Plague is an infectious disease that, despite affecting humans for more than 5000 years, has historically been the subject of limited drug development activity. In fact, the drugs currently used to treat plague have been approved based on experimental data alone–no major clinical trials have yet been completed that demonstrate the efficacy and safety of one treatment over another in humans. A major barrier to accomplishing this is that few research studies have taken place to date that can meaningfully inform the design of a clinical trial. We conducted this systematic review to gather and summarise all the existing information on the clinical profile of plague patients to understand whether sufficient information exists on which to design informative clinical trials that would provide clinicians and policy-makers with the information needed to make treatment decisions for patients.

This study however found that, based on the existing literature, there is insufficient data that can be used to develop methodologies for future trials. Either time must be invested in to collecting robust clinical data or innovative trial designs need to be identified that can simultaneously collect much-needed observational data while also evaluating much-needed interventions.

## Introduction

Plague is a zoonotic infectious disease caused by *Yersinia pestis*, Gram-negative bacteria most commonly found in small mammals and transmitted by fleas [1].

Transmission to humans typically occurs following a bite from an infected flea, contact with contaminated bodily fluids or tissues, or inhalation of respiratory droplets [1]. Once infected, human plague manifests in three main clinical forms: bubonic, pneumonic and septicaemic–the most common of which is bubonic plague, but the disease can also progress to include secondary clinical forms [1,2].

Other than characteristic swollen, painful lymph nodes–or "buboes"–symptoms of bubonic plague are often non-specific and can typically include fever, headache, chills and weakness [1,2]. Estimates of the case fatality rate (CFR) of bubonic plague are wide-ranging, extending from 24% to 60% overall [1,3], but a lower CFR of ~5% has been reported in treated cases and in research settings [4].

In 2018, the last year for which global data exist, 248 cases of plague were reported– 98% of which derive from just two countries, Madagascar and the Democratic Republic of Congo [5]. However, since 2013, cases have been consistently reported within a number of countries across Asia, North America, South America, and Africa and the global distribution of natural plague foci is thought to extend far beyond the countries that have recently reported cases [5,6]. Due to the disease's ability to lie dormant for years and even decades, [7] there is potential for cases to arise in locations in which plague has not been reported for a significant amount of time.

While several treatment regimens for plague are recommended in both national and international treatment guidelines, there are no robust clinical data to support the drugs that are

commonly used in clinical practice against plague. No pivotal randomised controlled trials have successfully taken place to generate sufficient evidence for plague treatment regimens. A number of drugs, such as streptomycin, ciprofloxacin and doxycycline [8], have been approved for the treatment of plague based on the U.S. Food and Drug Administration's (FDA) so-called 'Animal rule'–which permits the approval of drugs on the basis of well-controlled animal studies when human studies are neither ethical nor feasible [9]. Other drugs, which have not received FDA approval, such as gentamicin, are also commonly used to treat plague based on experimental data and clinical experience [10].

Some of these drugs however have significant drawbacks. Until recently, guidelines relied heavily on the use of aminoglycosides, with streptomycin in particular being the drug of choice for the treatment of plague in many settings [11,12–14]. While effective against Gram-negative bacterial infections, patients treated with aminoglycosides are at higher risk of experiencing ototoxic or nephrotoxic side effects than when treated with other drugs [15], requiring onerous clinical monitoring–which many low-resource health facilities in areas with the highest burden of plague are not equipped to carry out. Streptomycin also comes with a high cost and is currently in short supply on the global market. Other treatment regimens, such as those using doxycycline, are bacteriostatic, have only oral formulations available and therefore are not adapted for more severe forms of plague. Both aminoglycosides and tetracyclines are contraindicated in pregnancy [16].

Robust clinical studies are therefore required to identify safe and effective treatment regimens for plague. To date, only a small number of interventional trials have attempted to evaluate the efficacy of plague treatment regimens, two of which have published data [4,17]. Neither of these studies were able to generate sufficient evidence to demonstrate clinical benefit of their investigational drugs. This is in part due to the low numbers of patients enrolled in the trials–five and 65 patients were enrolled respectively–influenced by logistical challenges and the low numbers of cases reported in the areas in which the studies were conducted. The other challenge is methodological. One trial was designed as a non-randomised trial without a control group and the other was designed to recruit a small number of cases–for which the sample size estimation was based on data generated in a substantively different patient population–to assess a broadly-defined composite endpoint of cure or improvement in condition.

It is therefore clear that more work needs to be done to refine clinical trial methodologies for plague, keeping in mind the logistical challenges and the current low case numbers that might prevent the success of large-scale trials. Understanding the clinical characterisation of plague would be beneficial to overcoming some of the methodological challenges evident in the two studies referenced above, such as identifying viable, clinically relevant endpoints.

This review therefore aims to summarise the clinical profile of patients with bubonic plague–including clinical characteristics at baseline and post-treatment, and clinical outcomes–as described in published, peer-reviewed scientific literature.

## Methods

We conducted a systematic review to describe the clinical characteristics and outcomes of patients with suspected or confirmed bubonic plague from presentation to last recorded observation.

A search was conducted on bibliographic databases and clinical trial registries, including PubMED, Cochrane CENTRAL, clinicaltrials.gov, ISRCTN and the International Clinical Trials Registry Platform (ICTRP) for peer-reviewed publications describing the clinical characteristics of patients with bubonic plague. A supplementary search was conducted on JSTOR to identify older reports (pre-1970) of plague, although no formal search was conducted to obtain

these publications. A full list of search terms and filters that were used for each database and registry can be found in **S1 Text**.

To obtain data relating to the clinical characteristics of bubonic plague, all study designs were eligible for inclusion providing the publication contained individual patient data for adults or children of any age with suspected or confirmed bubonic plague, and described signs, symptoms and outcomes. There were no restrictions placed on language or publication year. Non-human, non-clinical, post-mortem and vaccine studies were excluded.

Two reviewers independently conducted screening in Rayyan [18] and was completed in two stages by reviewing titles and abstracts then reviewing full-text articles of remaining publications. Data were extracted by the first reviewer on to a standardised data capture form in Excel (**S1 Data**). A second reviewer verified the search and performed a quality control review on 30% of the data records. Any disagreement was discussed between the two reviewers and a third independent reviewer was involved if a disagreement remained unresolved.

Risk of bias was evaluated within the included studies by one reviewer using the Joanna Briggs Institute Critical Appraisal tools [19]. Case reports were assessed using the case reports tools, randomised controlled trials were assessed using the randomised controlled trials tool and non-randomised interventional studies were assessed using the quasi-experimental studies tool.

The protocol and data dictionary associated with this review can be accessed upon reasonable request to the corresponding author.

## Analysis

**The analysis was completed using R v.4.2.2..**   The screening and inclusion processes are summarised in the PRISMA flow diagram (**S1 Table**). A summary of all included studies is provided in a table detailing the study title, study type, country in which the study was conducted, number of bubonic plague cases in each study, the ratio of males to females, and the median and range of ages of the study population in years.

Two bar charts summarise the number of included studies per year per country, and the size of the patient population in the studies per year per country.

Demographic data were extracted from each study to summarise the ratio of males to females, and age (median and range) across the entire patient population of all included studies. The total number and percent of pregnant women and patients with comorbidities among the patient population is also provided.

The signs and symptoms are summarised for patients who received a confirmed clinical or laboratory diagnosis of bubonic plague and are reported at baseline and post-baseline timepoints. Baseline is defined at the first interaction the patient has with a health professional and post-baseline includes all timepoints that occur after the day of the initial interaction. Signs and symptoms are summarised as the number and percentage of patients in whom each sign or symptom was reported on at least one occasion. The denominator is based on the sum of the sample sizes of studies in which at least one patient reported the sign or symptom; in order to eliminate studies in which the sign or symptom may not have been assessed or captured during data collection, studies which do not report the sign or symptom for at least one patient have been excluded from the denominator calculation.

The number and percentage of patients receiving treatment or no treatment is provided. Treatment information is summarised according to antimicrobial class, as defined in a recent systematic review by the U.S. Centres for Disease Control and Prevention [20]. Aminoglycosides, tetracyclines, fluoroquinolones, sulphonamides, and amphenicols were considered to be "high-efficacy". All other antimicrobials were classified as "other antibiotics". The number and

percentage of deaths among those who received high-efficacy, other antibiotics or no treatment has been provided.

Patient outcomes are summarised as the number and percentage of deaths among the entire patient population, and according to the patient population in studies taking place in high-income countries and low- and middle-income countries. The outcome of the bubo–as present or not present–at the last recorded observation has been reported as the number and percentage of patients reporting the outcome among the patient population with a known bubo.

## Results

In total, 2023 publications were identified in the search (**Fig 1**). After removing duplicates, 1984 publications were screened for inclusion. Following title, abstract and full-text screening, 1897 records were excluded, resulting in the inclusion of 87 publications for data synthesis.

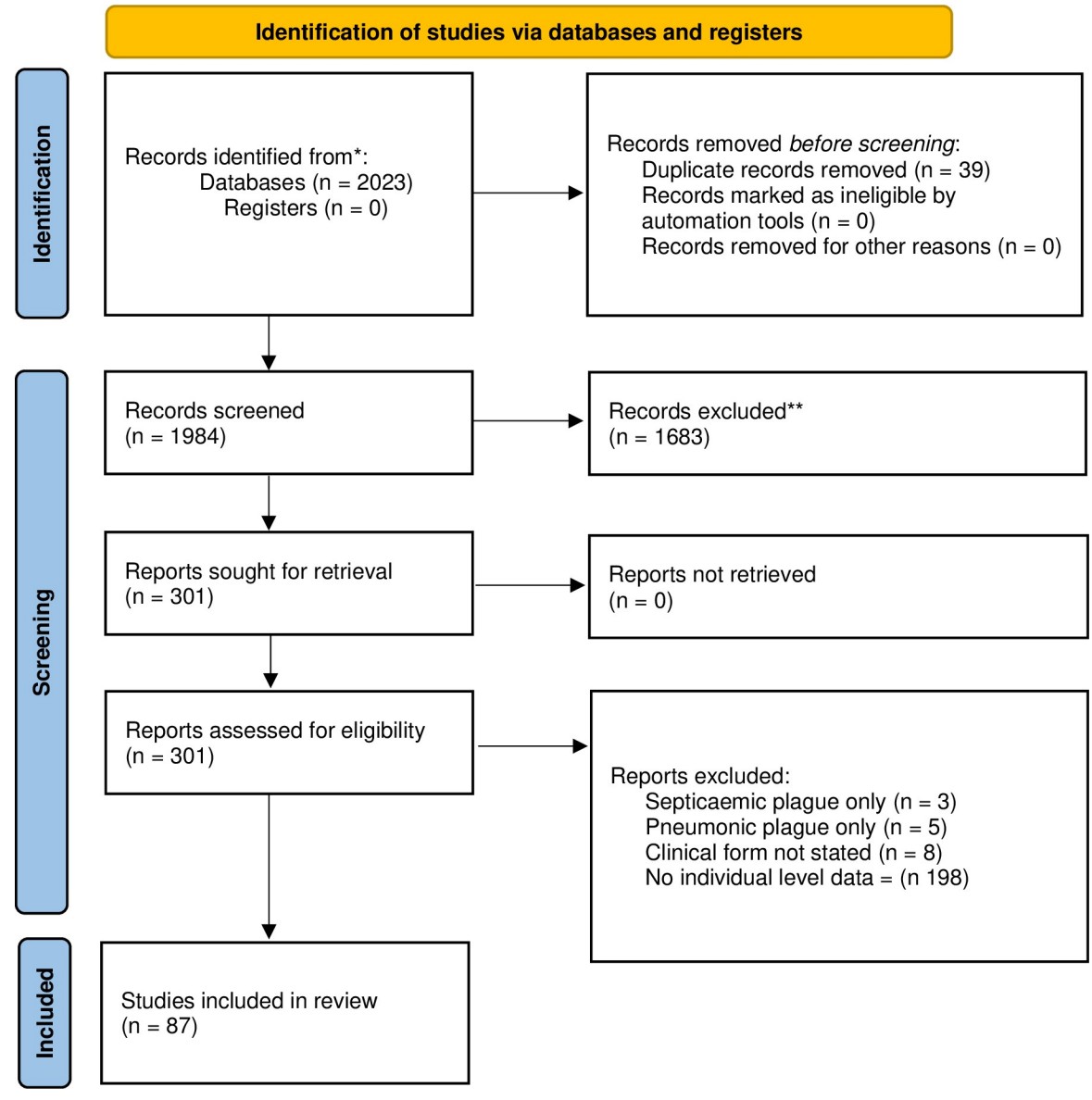

**Fig 1. PRISMA 2020 Flow Diagram [21].**

Of the included studies, 72 (83%) were case reports, 11 (13%) were cohort studies, four (4%) were interventional studies, of which one (1%) was randomised and three (3%) were non-randomised (**Table 1**).

There was a moderate risk of bias across all study types (**S2 Table** and **S1 Fig**). Case reports contained limited demographic, treatment and follow-up information, although baseline signs and symptoms, and diagnostic information was well-described. In interventional and cohort studies, outcomes were generally reliably described and assessed, although the completeness of follow-up information was often unclear.

All studies were published between 1902 and 2021 (**Fig 2A**).

Of the 87 included studies, 73 (84%) were conducted in the United States, three (3%) in the United Kingdom and Vietnam, two (2%) in Zambia, and 1 (1%) study each in Israel, Libya, Madagascar, Mongolia, Tanzania, and Uganda (**Fig 2A**).

These studies contain data on 1343 probable and confirmed cases of bubonic plague (including patients with secondary pneumonic or septicaemic plague). Of the included population, 870 (65%) derive from one study conducted in Madagascar, 307 (23%) from studies conducted in the United States, 76 (6%) from Vietnam, 65 (5%) from Tanzania, eight (1%) from Zambia, six (<1%) from each the United Kingdom, four (<1%) from Uganda, three (<1%) from Israel, and two (<1%) from Mongolia and Libya (**Fig 2B**).

However, over time the international distribution of included individuals has changed. From the 1960s to the 1980s reports of plague mainly derived from the United States; during this period, 209 (73%) individuals reported in this review were assessed in the United States (mostly in case reports of single patients or case series of small numbers of patients), and the remainder in Vietnam. Since the 1990s, the focus has shifted more towards the African region; during this time, 91% of cases included in this review were assessed in Madagascar, Tanzania, Zambia, Uganda and Libya, with the remainder assessed in the United States and Mongolia.

The study population consists of 769 (57%) males and 588 (43%) females with ages ranging from 0 to 82 years and a median age of 17 years. Among the population are five (<1%) patients who were pregnant at the time of their plague diagnosis. Comorbidities were reported for 17 (<1%) patients, including 13 (<1%) who tested positive for malaria. Other comorbidities include chronic cough, chronic renal insufficiency, idiopathic thrombocytopenia, and heart murmur.

## Reported signs and symptoms

**Baseline signs and symptoms.**   The median time from symptom onset to admission was 2 days, with a range of 0 to 18 days.

A diverse group of signs and symptoms were reported at baseline and post-baseline time-points for patients with confirmed bubonic plague within the articles included in this review (**S3 Table**).

The most commonly reported symptom at baseline was the presence of a bubo–an enlarged lymph node–which was recorded in 1216/1265 (96%) cases (**Table 1**). Of these cases, the median number of buboes reported per patient was one. Nineteen patients reported having multiple buboes, for whom the median number of buboes per patient was two. One patient was recorded as having 10 enlarged inguinal or suprapubic nodes, which were detected through the use of ultrasound. The location of the bubo was reported in 1090 (90%) of these patients. Buboes were most commonly reported in the inguinal area (590/1090, 54%), followed by the axillary (276/1090, 25%) and cervical areas (159/1090, 15%), and less frequently in other locations, such as in femoral and epitrochlear areas. Pain at the site of the bubo was recorded in 230/320 (72%) patients and bubo size was infrequently recorded– 19/88 (22%) publications

**Table 1.  Summary of included studies.**

| Study title *First author, Year* | Study type | Country | Bubonic plague cases reported, N | Sex, N male: N female | Age (years), Median[a] (range) |
|---|---|---|---|---|---|
| Two Cases Of Bubonic Plague Occurring On Board Ship *Barnett H. N., 1902* [22] | Case report(s) | United Kingdom | 2 | 2: 0 | NK |
| A case of bubonic plague on a vessel arriving in the Mersey *NK, 1905* [23] | Case report(s) | United Kingdom | 1 | 1: 0 | NK |
| Subacute plague in man due to ground squirrel infection *McCoy G. W., 1909* [24] | Case report(s) | United States | 1 | 1: 0 | 13 |
| Three Cases of Bubonic Plague Arising In England *Rendle-Short A., 1916* [25] | Case report(s) | United Kingdom | 3 | 3: 0 | 16 (10 to 23) |
| Streptomycin in Bubonic Plague *Haddad C. H., 1948* [26] | Interventional, non-randomised | Israel | 3 | 3: 0 | 20 (8 to 35) |
| Plague—New Mexico *United States Centres for Disease Control and Prevention, 1965* [27] | Case report(s) | United States | 1 | 1: 0 | 14 |
| Plague: Shasta County, California *United States Centres for Disease Control and Prevention, 1965* [28] | Case report(s) | United States | 1 | 1: 0 | 5 |
| Suspected Case of Imported Bubonic Plague *United States Centres for Disease Control and Prevention, 1966* [29] | Case report(s) | United States | 1 | 1: 0 | 21 |
| Plague—Arizona *United States Centres for Disease Control and Prevention, 1967* [30] | Case report(s) | United States | 1 | 1: 0 | 4 |
| Plague in San Diego *Connor J. D., 1968* [31] | Case report(s) | United States | 1 | 1: 0 | 3 |
| Presumptive Bubonic Plague—Denver, Colorado *United States Centres for Disease Control and Prevention, 1968* [32] | Case report(s) | United States | 1 | 0: 1 | 6 |
| Bubonic Plague Death—Lemhi County, Idaho *United States Centres for Disease Control and Prevention, 1968* [33] | Case report(s) | United States | 1 | 1: 0 | 32 |
| Plague Case—Navajo Reservation—Kayenta, Arizona *United States Centres for Disease Control and Prevention, 1968* [34] | Case report(s) | United States | 1 | 0: 1 | 8 |
| Plague—New Mexico *United States Centres for Disease Control and Prevention, 1969* [35] | Case report(s) | United States | 1 | 1: 0 | 3 |
| Bubonic plague in the Southwestern United States *Reed W. P., 1970* [36] | Case report(s) | United States | 25 | 15: 10 | 14 (2 to 74) |
| Bubonic Plague–California *United States Centres for Disease Control and Prevention, 1970* [37] | Case report(s) | United States | 1 | 1: 0 | 8 |
| Bubonic Plague–California *United States Centres for Disease Control and Prevention, 1970* | Case report(s) | United States | 1 | 1: 0 | 45 |
| Human Bubonic Plague—Cochiti, New Mexico *United States Centres for Disease Control and Prevention, 1970* [38] | Case report(s) | United States | 1 | 1: 0 | 39 |
| Bubonic Plague—Santa Fe, New Mexico *United States Centres for Disease Control and Prevention, 1970* [39] | Case report(s) | United States | 1 | 0: 1 | 20 |
| Plague—New Mexico *United States Centres for Disease Control and Prevention, 1970* [40] | Case report(s) | United States | 1 | 1: 0 | 13 |
| Plague—New Mexico *United States Centres for Disease Control and Prevention, 1970* [41] | Case report(s) | United States | 2 | 1: 1 | 24.5 (15 to 34) |
| Plague—California *United States Centres for Disease Control and Prevention, 1970* [42] | Case report(s) | United States | 1 | 0: 1 | 10 |
| Plague—Rio en Medio, New Mexico *United States Centres for Disease Control and Prevention, 1970* [43] | Case report(s) | United States | 1 | 0: 1 | 9 |
| Plague–New Mexico *United States Centres for Disease Control and Prevention, 1970* [44] | Case report(s) | United States | 1 | 1: 1 | 11.5 (7 to 16) |
| Human Bubonic Plague–Oregon *United States Centres for Disease Control and Prevention, 1971* [45] | Case report(s) | United States | 1 | 1: 0 | 10 |
| Human Bubonic Plague–New Mexico *United States Centres for Disease Control and Prevention, 1971* [46] | Case report(s) | United States | 1 | 0: 1 | 26 |

*(Continued)*

**Table 1.** (Continued)

| Study title First author, Year | Study type | Country | Bubonic plague cases reported, N | Sex, N male: N female | Age (years), Median[a] (range) |
|---|---|---|---|---|---|
| Clinical Features of Plague in the United States: the 1969–1970 Epidemic<br>Palmer D. L., 1971 [47] | Cohort | United States | 19 | 12: 7 | 25.5 (2 to 49) |
| Human Bubonic Plague—Coconino County Colorado<br>United States Centres for Disease Control and Prevention, 1972 [48] | Case report(s) | United States | 1 | 1: 0 | 19 |
| Human Bubonic Plague–Arizona<br>United States Centres for Disease Control and Prevention, 1973 [49] | Case report(s) | United States | 1 | 0: 1 | 9 |
| Co-trimoxazole in Bubonic Plague<br>Nguyen-Van-Ai, 1973 [50] | Interventional, non-randomised | Vietnam | 12 | 4: 8 | 38.5 (6 to 63) |
| Human Bubonic Plague–New Mexico<br>United States Centres for Disease Control and Prevention, 1974 [51] | Case report(s) | United States | 1 | 0: 1 | 12 |
| Human Plague–New Mexico<br>United States Centres for Disease Control and Prevention, 1974 [52] | Case report(s) | United States | 1 | 1: 0 | 19 |
| Human Plague–New Mexico, Utah<br>United States Centres for Disease Control and Prevention, 1974 [53] | Case report(s) | United States | 1 | 1: 0 | 5 |
| Yersinia pestis Infection in Vietnam. I. Clinical and Hematologic Aspects<br>Butler T., 1974 [54] | Cohort | Vietnam | 22 | 17: 5 | 16 (4 to 67) |
| Plague and the gallium scan<br>Stahly T. L., 1975 [55] | Case report(s) | United States | 1 | 1: 0 | 11 |
| Fatal Human Plague–California<br>United States Centres for Disease Control and Prevention, 1975 [56] | Case report(s) | United States | 1 | 0: 1 | 1 |
| Bubonic Plague–Arizona<br>United States Centres for Disease Control and Prevention, 1975 [57] | Case report(s) | United States | 2 | 0: 2 | 17 (3 to 31) |
| Plague in Humans–New Mexico<br>United States Centres for Disease Control and Prevention, 1975 [58] | Case report(s) | United States | 6 | 0: 6 | 10.5 (3 to 28) |
| Human Plague Case—Bernalillo County, New Mexico<br>United States Centres for Disease Control and Prevention, 1975 [59] | Case report(s) | United States | 1 | 1: 0 | 11 |
| Bubonic Plague from Exposure to a Rabbit: A Documented Case, and a Review of Rabbit-Associated Plague Cases in The United States<br>Von Reyn C. F., 1976 [60] | Case report(s) | United States | 1 | 0: 1 | 62 |
| Human Plague—Arizona, California, New Mexico<br>United States Centres for Disease Control and Prevention, 1976 [61] | Case report(s) | United States | 3 | 2: 1 | 45 (15 to 63) |
| Yersinia pestis Infection in Vietnam. II; Quantitative Blood Cultures and Detection of Endotoxin in the Cerebrospinal Fluid of Patients with Meningitis<br>Butler T, 1976 [62] | Cohort | Vietnam | 42 | 20: 22 | 15 (3 to 70) |
| Plague and pregnancy. A case report<br>Mann J, 1977 [63] | Case report(s) | United States | 1 | 0: 1 | 28 |
| Epidemiological and clinical features of an outbreak of bubonic plague in New Mexico<br>Von Reyn C. F., 1977 [64] | Cohort | United States | 7 | 3: 4 | 19 (5 to 62) |
| Plague—Arizona, Colorado, New Mexico<br>United States Centres for Disease Control and Prevention, 1977 [65] | Case report(s) | United States | 4 | 3: 1 | 30.5 (3 to 43) |
| Plague–United States<br>United States Centres for Disease Control and Prevention, 1977 [66] | Case report(s) | United States | 1 | 1: 0 | 6 |
| Plague—Arizona, California, New Mexico<br>United States Centres for Disease Control and Prevention, 1978 [67] | Case report(s) | United States | 1 | 1: 0 | 14 |
| Plague in the United States: the "black death" is still alive<br>Hoffman S. L., 1980 [68] | Case report(s) | United States | 1 | 1: 0 | 55 |
| Plague—United States<br>United States Centres for Disease Control and Prevention, 1980 [69] | Case report(s) | United States | 9 | 6: 3 | 8 (4 to 55) |

(Continued)

**Table 1.** (Continued)

| Study title *First author, Year* | Study type | Country | Bubonic plague cases reported, N | Sex, N male: N female | Age (years), Median[a] (range) |
|---|---|---|---|---|---|
| Human Plague–Texas, New Mexico<br>*United States Centres for Disease Control and Prevention, 1981* [70] | Case report(s) | United States | 1 | 1: 0 | 25 |
| Human plague associated with domestic cats—California, Colorado<br>*United States Centres for Disease Control and Prevention, 1981* [71] | Case report(s) | United States | 1 | 1: 0 | 49 |
| Peripatetic Plague<br>*Mann J., 1982* [72] | Case report(s) | United States | 1 | 0: 1 | 16 |
| Febrile lymphadenitis in the American West<br>*Mann J., 1982* [73] | Case report(s) | United States | 1 | 1: 0 | 5 |
| Plague—South Carolina<br>*United States Centres for Disease Control and Prevention, 1983* [74] | Case report(s) | United States | 1 | 0: 1 | 13 |
| Plague in the United States 1982<br>*Barnes A. M., 1983* [75] | Cohort | United States | 19 | 11: 8 | 20 (4 to 78) |
| Plague Pneumonia–California<br>*United States Centres for Disease Control and Prevention, 1984* [76] | Case report(s) | United States | 1 | 1: 0 | 35 |
| Winter Plague—Colorado, Washington, Texas, 1983–1984<br>*United States Centres for Disease Control and Prevention, 1984* [77] | Case report(s) | United States | 3 | 3: 0 | 38.5 (29 to 48) |
| Human Bubonic Plague Transmitted by a Domestic Cat Scratch<br>*Weniger B. G., 1984* [78] | Case report(s) | United States | 1 | 0: 1 | 10 |
| Nineteen cases of plague in Arizona. A spectrum including ecthyma gangrenosum due to plague and plague in pregnancy<br>*Welty T. K., 1985* [79] | Cohort | United States | 19 | 6: 13 | 25 (2 to 78) |
| Multiple lung cavities in a 12-year-old girl with bubonic plague, sepsis, and secondary pneumonia<br>*Florman A. L., 1986* [80] | Case report(s) | United States | 1 | 0: 1 | 12 |
| Plague masquerading as gastrointestinal illness<br>*Hull H. F., 1986* [81] | Cohort | United States | 47 | NK: NK | 61 (13 to 71) |
| Plague in a pregnant patient<br>*Wong T. W., 1986* [82] | Case report(s) | United States | 1 | 0: 1 | 25 |
| Plague meningitis—a retrospective analysis of cases reported in the United States, 1970–1979<br>*Becker T. M., 1987* [83] | Case report(s) | United States | 2 | 1: 1 | 10.5 (10 to 11) |
| Imaging in plague<br>*Moreno A. J., 1987* [84] | Case report(s) | United States | 1 | 1: 0 | 8 |
| Human Plague—United States, 1988<br>*United States Centres for Disease Control and Prevention, 1988* [85] | Case report(s) | United States | 3 | 3: 0 | 41 (19 to 82) |
| Imported bubonic plague—District of Columbia<br>*United States Centres for Disease Control and Prevention, 1990* [86] | Case report(s) | United States | 1 | 0: 1 | 47 |
| Bubonic plague in a child presenting with fever and altered mental status<br>*Migden D., 1990* [87] | Case report(s) | United States | 1 | 1: 0 | 8 |
| Plague in New Mexico<br>*Owens C., 1990* [88] | Case report(s) | United States | 1 | 1: 0 | 47 |
| Plague—A clinical review of 27 cases<br>*Crook L. D., 1992* [89] | Cohort | United States | 27 | 15: 12 | 41 (2 to 80) |
| An Outbreak of Plague in Northwestern Province, Zambia<br>*McClean K. L., 1995* [90] | Case report(s) | Zambia | 1 | 1: 0 | 23 |
| Fatal human plague—Arizona and Colorado, 1996<br>*United States Centres for Disease Control and Prevention, 1997* [91] | Case report(s) | United States | 2 | 1: 1 | 17 (16 to 18) |
| Current epidemiology of human plague in Madagascar<br>*Chanteau S., 2000* [92] | Cohort | Madagascar | 917 | 515: 402 | 15[b] |
| Cases of cat-associated human plague in the Western US, 1977–1998<br>*Gage K. L., 2000* [93] | Case report(s) | United States | 17 | 11: 6 | 28 (6 to 58) |

(*Continued*)

**Table 1.** (Continued)

| Study title *First author, Year* | Study type | Country | Bubonic plague cases reported, N | Sex, N male: N female | Age (years), Median[a] (range) |
|---|---|---|---|---|---|
| Imported plague—New York City, 2002 *United States Centres for Disease Control and Prevention, 2003* [94] | Case report(s) | United States | 2 | 1: 1 | 50 (47 to 53) |
| Painful lymphadenopathy and fulminant sepsis in a previously healthy 16-year-old girl *Chmura K., 2003* [95] | Case report(s) | United States | 2 | 0: 1 | 16 |
| Gentamicin and Tetracyclines for the Treatment of Human Plague: Review of 75 cases in New Mexico, 1985–1999 *Boulanger L. L., 2004* [96] | Cohort | United States | 50 | 31: 19 | NK |
| Human plague—four states, 2006 *United States Centres for Disease Control and Prevention, 2006* [97] | Case report(s) | United States | 4 | 2: 2 | 34 (28 to 43) |
| Treatment of Plague with Gentamicin or Doxycycline in a Randomized Clinical Trial in Tanzania *Mwengee W., 2006* [4] | Interventional, randomised controlled trial | Tanzania | 65 | 41: 24 | 13 (0 to 65) |
| Notes from the field: two cases of human plague—Oregon, 2010 *United States Centres for Disease Control and Prevention, 2011* [98] | Case report(s) | United States | 2 | NK: NK | 29.5 (17 to 42) |
| Misidentification of Yersinia pestis by Automated Systems, Resulting in Delayed Diagnoses of Human Plague Infections—Oregon and New Mexico, 2010–2011 *Tourdjman M., 2012* [99] | Case report(s) | United States | 2 | 1: 1 | 37.5 (17 to 58) |
| Plague Outbreak in Libya, 2009, Unrelated to Plague in Algeria *Cabanel N., 2013* [100] | Cohort | Libya | 2 | 1: 1 | 19 (14 to 24) |
| Outbreak of Plague in a High Malaria Endemic Region—Nyimba District, Zambia, March–May 2015 *Sinyange N., 2016* [101] | Cohort | Zambia | 7 | NK: NK | 8 (3 to 18) |
| Successful Treatment of Human Plague with Oral Ciprofloxacin *Apangu T., 2017* [17] | Interventional, non-randomised | Uganda | 4 | 1: 3 | 31 (10 to 52) |
| Case report *Lazet K., 2018* [102] | Case report(s) | United States | 1 | 0: 1 | 33 |
| Human case of bubonic plague resulting from the bite of a wild Gunnison's prairie dog during translocation from a plague endemic area *Melman S. D., 2018* [103] | Case report(s) | United States | 1 | 1: 0 | 66 |
| Two fatal cases of plague after consumption of raw marmot organs *Kehrmann J., 2020* [104] | Case report(s) | Mongolia | 2 | 1: 1 | 37.5 (37 to 38) |
| Delays in Identification and Treatment of a Case of Septicemic Plague—Navajo County, Arizona, 2020 *Dale A. P., 2021* [105] | Case report(s) | United States | 1 | 1: 0 | 67 |

[a] Median and range only given where article includes >1 case of bubonic plague

[b] Only median age available

noted the size of at least one patient's bubo. The median recorded bubo size was 30mm, with a range of 1mm to 150mm. The method of bubo measurement was reported only in one case report describing one patient, in which ultrasound was used.

Following the presence of a bubo, fever was the most frequently reported systemic feature at baseline, present in 1004/1279 (78%) patients, and for which the median recorded temperature was 39.5˚C (range: 36˚C to 41.5˚C) (**Table 2**). Headache was present in 127/244 (52%) patients, followed by altered mental status (33/84, 39%), chills (75/195, 38%) and malaise (22/68, 32%).

Fewer patients presented with clinical signs and symptoms indicative of severe illness at baseline, such as seizure (4/34, 12%), septic shock (5/59, 8%), and respiratory distress, arrest or failure (3/5, 60%).

**Fig 2a – Number of studies included by country and by year**

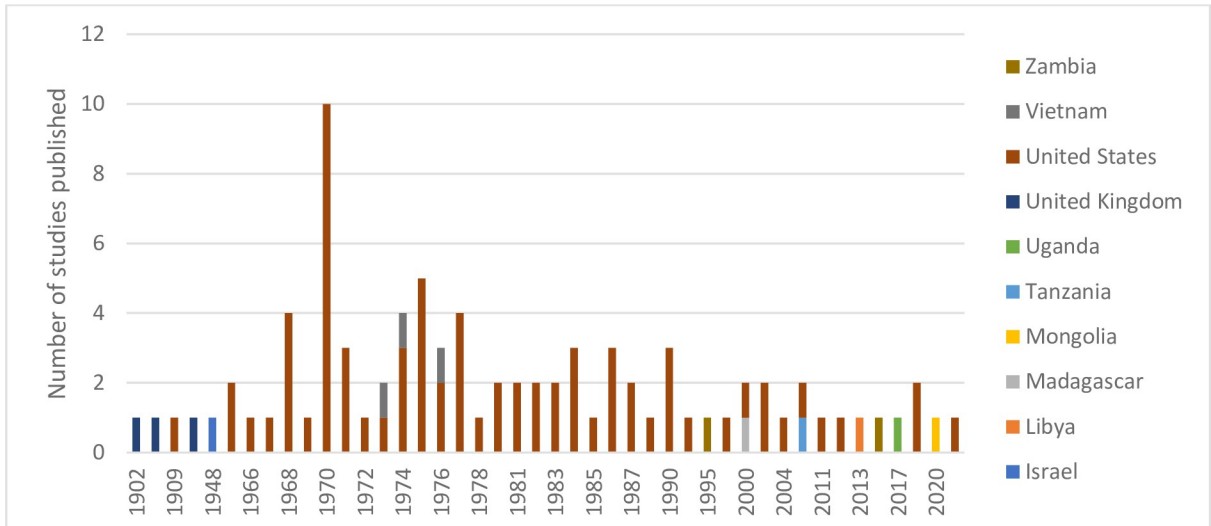

**Fig 2b – Number of patients reported by country and by year**

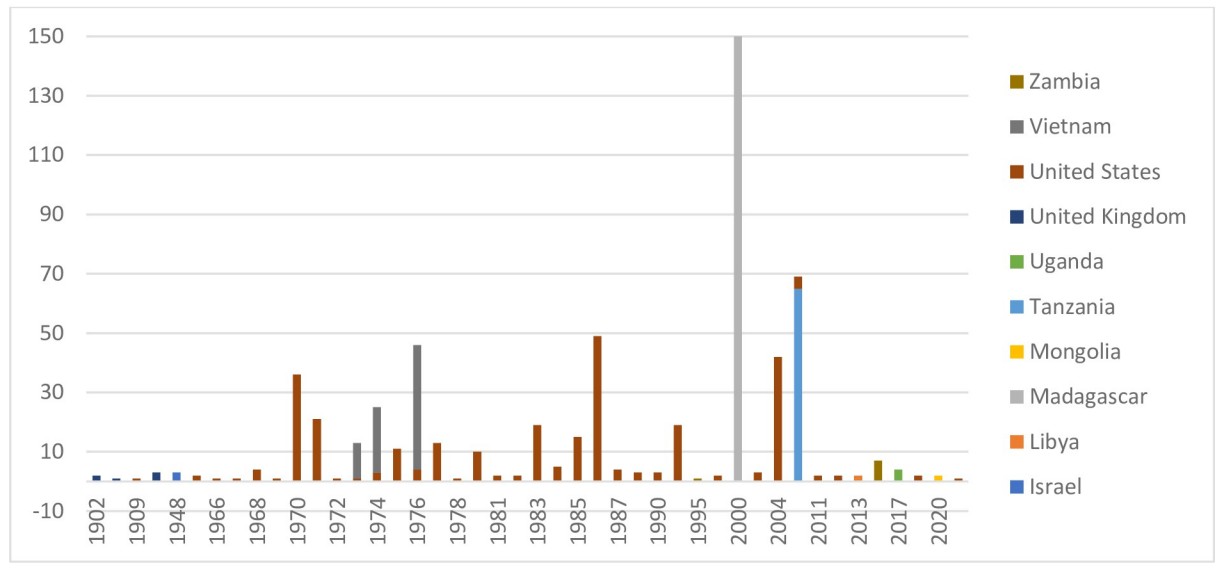

**Fig 2. A) Number of studies included by country and by year; B) Number of patients reported by country and by year.**

All other signs and symptoms are reported in **S3 Table**.

**Post-baseline signs and symptoms.** Reporting of signs and symptoms decreased following the baseline assessment. Patients were followed up for a median of 7 days from admission (IQR 2 days to 20 days).

Reports of fever and headache persisted each in 43% of patients, and there were infrequent reports of other common symptoms at baseline, such as malaise, fatigue, abdominal pain, vomiting, cough, nausea, hypotension, and diarrhoea. While altered mental status and chills were reported in relatively high proportions of patients at baseline, no reports of these symptoms were made post-baseline.

**Table 2. Reported signs and symptoms at baseline and post-baseline, N (%) patients.**

| | N (%) patients for whom symptom reported | |
| --- | --- | --- |
| | **Baseline** | **Post-baseline** |
| **Fever** | 1004/1279 (78%) | 52/120 (43%) |
| *Temperature, °C (median, range)* | *39.5, 36 to 41.5* | - |
| **Headache** | 127/244 (52%) | 3/69 (43%) |
| **Altered mental status** | 33/84 (39%) | - |
| **Chills** | 75/195 (38%) | - |
| **Malaise** | 22/68 (32%) | 1/1 (100%) |
| **Fatigue** | 6/22 (27%) | 1/3 (33%) |
| **Myalgia** | 54/204 (26%) | - |
| **Abdominal pain** | 48/194 (25%) | 2/2 (100%) |
| **Vomiting** | 63/261 (24%) | 5/70 (7%) |
| **Sore throat** | 17/73 (23%) | - |
| **Cough** | 37/167 (22%) | 12/115 (10%) |
| **Nausea** | 41/191 (21%) | 1/65 (2%) |
| **Hypotension** | 15/87 (17%) | 11/85 (13%) |
| **Diarrhoea** | 22/147 (15%) | 3/67 (4%) |
| **Seizure** | 4/34 (12%) | 2/66 (3%) |
| **Septic shock** | 5/59 (8%) | 2/4 (50%) |
| **Respiratory distress, arrest or failure** | 3/5 (60%) | 8/65 (12%) |
| **Disseminated intravascular coagulation** | 1/1 (100%) | 7/41 (17%) |
| **Bubo** | 1216/1265 (96%) | 61/93 (64%) |
| *Axillary* | *276/1090 (25%)* | - |
| *Cervical* | *159/1090 (15%)* | - |
| *Inguinal* | *590/1090 (54%)* | - |
| *Other* | *65/1090 (6%)* | - |
| *Pain at bubo site* | *230/320 (72%)* | - |
| *Size, mm (median, range)* | *30, 1 to 150* | - |

Only 19/88 (22%) articles provided information about the presence of the bubo at any time-point post-baseline for at least one patient in the study's cohort. Within this population, a persistent bubo was recorded for 61/95 (64%) patient. A repeated bubo measurement is described in two case reports, in which one repeated the measurement at 3 days post-baseline and demonstrated increasing inguinal lymphadenopathy, which subsequently reduced to 10mm 26 days post-baseline–a reduction of 25mm in total; and the other repeated the measurement at 29 days post-baseline and demonstrated a reduction in size to 5mm–a decrease of 15mm from the baseline measurement–from baseline in an axillary bubo which fully resolved at 39 days post-baseline [78].

However, the frequency of some signs and symptoms indicative of severe illness increased. After the baseline assessment, respiratory distress, arrest or failure, for example, was recorded in 8/65 (12%) patients and disseminated intravascular coagulation was recorded in 7/41 (17%) patients.

## Outcome

An outcome was recorded for 1300/1343 (97%) patients. At the time of the last recorded observation (median: 7 days; range: 0 to 137 days), 1090/1343 (81%) had fully recovered and death was reported in 208/1343 (15%) patients (**Table 3**); however, the case fatality ratio but varied

**Table 3. Summary of patient outcomes.**

| Outcome | All patients | HIC patients | LMIC patients |
|---|---|---|---|
| Death, n/N (%) | 208/1343 (15%) | 32/316 (10%) | 176/1027 (17%) |
| Time (days) from enrolment to death, median (range) | 1 (0 to 16) | 1 (0 to 16) | 2 (1 to 5) |
| Fully recovered at last reported observation | 1076/1343 (80%) | 227/316 (72%) | 849/1027 (83%) |
| Time (days) to defervescence, median (range) | 3.5 (0 to 21) | 4 (0 to 21) | 2 (0 to 4) |
| Outcome unknown | 45/1343 (3%) | 57/316 (18%) | 2/1027 (<1%) |

from 10% for cases reported in high-income countries (HICs) to 17% in low- and middle-income countries (LMICs). The median time to death was 1 day in HICs and 2 days in LMICs, ranging from 0 to 16 days.

A bubo outcome was reported in 93/1279 (7%) cases in whom a bubo was reported at baseline. In 32/93 (34%) patients the bubo had completely disappeared at the last reported observation and in 61/93 (66%) cases the bubo was still present at the time of the last reported observation, which ranged from 0 to 137 days post-admission.

## Treatment

Information on therapeutic intervention–or no intervention where specifically indicated–was available for 306/1343 (23%) patients, of whom 271/306 (89%) received a high-efficacy antimicrobial (with or without other antibiotics), 24/306 (8%) received only other antibiotics and 11/306 (4%) received no antibiotics (**Table 4**).

Streptomycin was the most frequently administered high-efficacy plague therapeutic that was administered–received by 141/306 (46%) patients either alone or in combination with other drugs–followed by gentamicin, tetracycline and doxycycline received by 91/306 (30%), 85/306 (28%), and 42/306 (14%) patients, respectively. All other drugs were administered in small numbers of patients.

Of those who received a high-efficacy antimicrobial at any time following initial presentation, 15/271 (6%) died. Of the 24 patients who received only other antibiotics, 6 (25%) died and of the 11 patients who received no antibiotic treatment, 5 (45%) died.

## Discussion

This systematic review was designed to summarise the clinical profile of patients with bubonic plague, as described in academic literature, with the objective of clarifying the endpoint(s) that could be used as the primary efficacy endpoint in therapeutic trials.

While this study has captured data on the signs, symptoms and outcomes of substantial number of confirmed cases of plague, it is limited to published studies written in English. There are several other studies, written in other languages, from which data have not been extracted. It must be acknowledged that studies describing cases in China, where plague cases have been well-documented [106], are not included in this review and only one study describing two cases from Mongolia has been included.

Overall the CFR that occurred across the patient population included in this review (15%) indicates that mortality might not be a suitable primary endpoint for a clinical trial–a future trial using a single mortality endpoint would likely require sample sizes that may not be attainable to detect significant treatment effects between arms. A rough sample size estimation designed to detect a 50% reduction in mortality with a significance level of 5%, 80% power and 10% lost to follow-up would yield a target sample size of 203 patients per arm, which might require several years to complete, or not be attainable at all.

**Table 4. Summary of treatments received by number of patients and number of deaths.**

| | Treatment with high-efficacy antibiotic | | | | | | | | | | | Other antibiotic treatment | No antibiotic treatment |
|---|---|---|---|---|---|---|---|---|---|---|---|---|---|
| | Streptomycin | Gentamicin | Levofloxacin | Doxycycline | Ciprofloxacin | Tetracycline | Chloramphenicol | Amikacin | Tobramycin | Trimethoprim-Sulfamethoxazole | Total | | |
| n (%) patients N = 306 | 141 (46%) | 91 (30%) | 1 (<1%) | 42 (14%) | 8 (3%) | 85 (28%) | 5 (2%) | 1 (<1%) | 1 (<1%) | 7 (2%) | 271 (89%) | 24 (8%) | 11 (4%) |
| N deaths (CFR) | 6 (4%) | 11 (12%) | 0 (0%) | 2 (5%) | 0 (0%) | 3 (4%) | 5 (7%) | 0 (0%) | 0 (0%) | 0 (0%) | 15 (6%) | 6 (25%) | 5 (45%) |

The data on CFR collected in the scope of this review–which varies according to context and type of treatment received by the patient–may however not be representative of bubonic plague in the 21$^{st}$ century. The majority of included studies are case reports which derive from the United States between the 1960s and 1980s and represent a large proportion of the patient population included in this review. In 2018 however–the most recent year for which global data are available– 80% of the global cases of plague derive from Madagascar, and a further 18% from the Democratic Republic of Congo. These are two countries in which healthcare availability and accessibility may be substantially different to that of the United States, which will likely influence patterns in disease presentation and patient outcomes. In Madagascar, the CFR for patients with bubonic plague between 1998 and 2016 was reported to be 15% [107]. The study reporting this figure however cited several limitations in that diagnostic testing evolved during the course of the study–meaning some earlier cases were potentially missed due to the sub-optimal sensitivity of older testing methods–and that samples from regions in Madagascar that had historically infrequently reported cases of plague were not consistently available. The CFR was higher at 24% during a large, atypical urban plague outbreak that occurred in 2017 [3]. Uncertainty therefore still remains around the true incidence and CFR for plague in the country with the greatest disease burden.

As survival or mortality may not be feasibly used as the single primary efficacy endpoint in a trial for bubonic plague, alternative or additional clinically-relevant outcomes must be identified. The only randomised controlled trial included in this review used "cure or improvement in condition" as its primary endpoint–a composite outcome measure consisting of 1) fever resolution; 2) resolution of bubo pain and swelling; 3) and recovery from pneumonia or any other symptom of plague present at baseline [4]. An ongoing trial in Madagascar also uses a composite endpoint of therapeutic response defined as 1) the patient being alive; 2) resolution of fever; 3) a 25% decrease in bubo size; 4) alternative plague treatment not received; 5) no decision to continue treatment beyond day 10 [108]. Composite primary endpoints such as these provide a useful solution for diseases where low numbers of individual events would prevent the detection of significant differences between arms in a trial with a single endpoint. For example, by identifying several outcomes of interest, the number of outcome events increases, therefore making it more likely that an effect can be detected with a reasonable level of confidence.

Composite endpoints however also have the potential to introduce uncertainty around any discernible differences that are detected and make the interpretation of the trial results challenging. For example, the clinical relevance of the individual criteria that make up the above-referenced composite endpoints is difficult to discern. Fever resolution, for example, was reported only for a small proportion of patients (14%) included in this review, and although the event rate– 161/186 (87%)–was high, its relationship with a patient's overall clinical status is unclear in the literature as limited longitudinal data are available. The same can be said of the persistence or size of the bubo.

The breadth of the events included in the composite endpoint also create challenges for the accurate interpretation of the trial's results. The events range from resolution of fever to resolution of pneumonia or any other symptom of plague. It is clear that the severity and clinical importance of these events are vastly different. However, they are weighted equally in this composite endpoint making it difficult to understand the true efficacy of the drug and how it works in this patient population. In the event that Patient A presents with only a fever at baseline and Patient B presents with pneumonia, fever and a bubo, can the recovery of both patients represent the true efficacy of the drug?

While the urgency to provide results for plague, a disease with little evidence to support current treatment recommendations, is understandable, critically, the relationship between

individual signs, symptoms and clinical status–as determined by both physical indicators of a patient's condition and laboratory testing for presence of *Y. pestis* in blood and/or bubo site–must be established before implementing a primary outcome measure of this nature in a clinical trial.

Given the existing data, however, it is not currently possible to identify a single or composite endpoint for plague, for which clinical relevance has been established and which would generate an achievable sample size. It is clear that robust longitudinal data from an observational study would provide critical information about the overall clinical course of plague illness and illuminate the important outcomes that could be used in either a single or composite endpoint.

However, this will take a substantial amount of time–which could be better spent by taking a more pragmatic approach to a trial. An approach recently proposed for Lassa fever, which exhibits similar challenges in endpoint definition, is to embed an observational cohort within the trial framework to collect data on patient outcomes [109]. The proposed design involves powering the trial for a mortality endpoint, and in parallel identifying and collecting data on several important outcomes of interest during the first year of the trial. An interim feasibility analysis would be undertaken to determine whether the frequencies of the events are sufficient to generate an achievable sample size and, if so, the trial would amend its primary endpoint. While this design has not been implemented in a trial to date, it may provide a useful path forward for diseases such as plague where little comparable data have been generated and uncertainty surrounds outcome frequencies. This framework may provide a cost-saving and efficient approach to trials for diseases like plague that could minimise delays to testing new drugs and reduce costs associated with implementing both observational studies and trials sequentially.

Overall, this systematic review demonstrates the restrictions that limited disease characterisation places on clinical trials for rare infectious diseases such as plague. The limited data that exist on the clinical course of plague and patient outcomes, not only impact the definition of trial endpoints but has the knock-on effect of challenging the interpretation of a trial's results. For this reason and despite interventional trials for plague having taken place, questions around optimal treatment for plague persist. Options however exist–including those for pragmatic and innovative trial designs. A bigger challenge may however be to catalyse R&D interest in a disease for which there is limited money to be made in the pharmaceutical industry.

## Supporting information

**S1 Text. Search strategy.**
(DOCX)

**S1 Table. PRISMA checklist.**
(DOCX)

**S2 Table. Risk of bias assessments.**
(DOCX)

**S3 Table. Other reported signs and symptoms at baseline and post-baseline.**
(DOCX)

**S1 Fig. Summary of risk of bias assessments.**
(TIF)

**S1 Data. Full dataset.**
(XLSX)

## Acknowledgments

The authors would like to thank Eli Harris for guidance and support to develop the search strategy in this review.

## Author Contributions

**Conceptualization:** Josephine Bourner, Alex Salam, Rindra Randremanana, Piero Olliaro.

**Data curation:** Josephine Bourner, Lovarivelo Andriamarohasina, Nzelle Delphine Kayem.

**Formal analysis:** Josephine Bourner.

**Investigation:** Josephine Bourner, Lovarivelo Andriamarohasina, Nzelle Delphine Kayem.

**Methodology:** Josephine Bourner, Alex Salam, Nzelle Delphine Kayem, Piero Olliaro.

**Project administration:** Josephine Bourner.

**Resources:** Rindra Randremanana, Piero Olliaro.

**Supervision:** Alex Salam, Rindra Randremanana, Piero Olliaro.

**Validation:** Lovarivelo Andriamarohasina.

**Visualization:** Josephine Bourner.

**Writing – original draft:** Josephine Bourner.

**Writing – review & editing:** Josephine Bourner, Alex Salam, Rindra Randremanana, Piero Olliaro.

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
