## [Decision Letter · Decision Letter 0]

27 Aug 2023

Dear Ms Bourner,

Thank you very much for submitting your manuscript "A systematic review of the clinical profile of patients with bubonic plague and the outcome measures used in research settings" for consideration at PLOS Neglected Tropical Diseases. As with all papers reviewed by the journal, your manuscript was reviewed by members of the editorial board and by several independent reviewers. In light of the reviews (below this email), we would like to invite the resubmission of a significantly-revised version that takes into account the reviewers' comments. 

Please, pay attention on comments from Reviewer 3 and try to change the presentation of the data from large Tables to graphical format when it is possible.

We cannot make any decision about publication until we have seen the revised manuscript and your response to the reviewers' comments. Your revised manuscript is also likely to be sent to reviewers for further evaluation.

Sincerely,

Vladimir L. Motin, Ph.D.

Guest Editor

Stuart Blacksell

Section Editor

Please, pay attention on comments from Reviewer 3 and try to change the presentation of the data from large Tables to graphical format when it is possible.

Reviewer's Responses to Questions

**Key Review Criteria Required for Acceptance?**

**Methods**

-Are the objectives of the study clearly articulated with a clear testable hypothesis stated?

-Is the study design appropriate to address the stated objectives?

-Is the population clearly described and appropriate for the hypothesis being tested?

-Is the sample size sufficient to ensure adequate power to address the hypothesis being tested?

-Were correct statistical analysis used to support conclusions?

-Are there concerns about ethical or regulatory requirements being met?

Reviewer #1: (No Response)

Reviewer #2: The objectives of the study are clear: the authors aim to identify endpoints that could be used as the primary efficacy endpoint in therapeutic trials on bubonic plague.

The study design seems appropriate. The criteria used to include studies are clear. The large number of included studies provides informative data on signs and symptoms.

Reviewer #3: This study was presented as a hypothesis-generating study, with a clearly stated goal to catalog the available data on plague patients, their disease course, and response to treatment. The study design and inclusion of data considered the relatively sparse amount of plague cases, which also justified the analysis. The authors were thoughtful in the inclusion criteria for the analysis. The population was significantly weighted to a single recent large outbreak in Madagascar; in addition the majority of the remaining cases were from the US. These factors may have reduced the study power, but also are somewhat out of the control of the authors.

**Results**

-Does the analysis presented match the analysis plan?

-Are the results clearly and completely presented?

-Are the figures (Tables, Images) of sufficient quality for clarity?

Reviewer #1: (No Response)

Reviewer #2: Presented analysis match the analysis plan.

Results are clearly and completely presented.

Figures and tables are clear. However, I noted some inconsistencies:

 - Table 3 should be revised:

Presentation of the results is not concordant with the number of reported outcomes. The authors should choose between to present the number of death and full recovery among the 1300 reported outcomes and to present the distribution of the outcomes among the 1343 patients and in this case, precise the number of unknown outcomes.

It will be worthy to indicate in table 3, the % of deaths observed in high-income countries and in low-and middle-income countries and the sex ratio of deaths in these 2 categories.

Persistent bubos already are mentioned in table 2 and should be removed from table 3

- There is a discordance in the number of patients with information on intervention. Lines 266-268:information was available for 305 patients, but 271 +24 +11 =306. The same discordance is present also in table 4.

Reviewer #3: The results were presented in tables, with one data figure that depicted where and when the included cases were drawn. It would be easier for the reader if the authors provided figures that supported the conclusions that could be drawn from the very large tables. For example, the authors could display graphically the most common outcomes reported; the authors could display in a graph, a comparison between the outcomes in the US compared to Madagascar, since these were the two largest groups; the authors could compare outcomes over time in all groups using a graph. I think any or all of these would provide clarity for the data that is presented, and might even lead to additional provocative conclusions.

**Conclusions**

-Are the conclusions supported by the data presented?

-Are the limitations of analysis clearly described?

-Do the authors discuss how these data can be helpful to advance our understanding of the topic under study?

-Is public health relevance addressed?

Reviewer #1: (No Response)

Reviewer #2: The conclusions are well supported by the data

the bias are presented in the results section

The discussion is interesting and may be helpful to advance in therapeutic trial for bubonic plague

Reviewer #3: The conclusions appear to be supported, but clarity could be improved by the presentation of the data and prevailing hypothesis on endpoints (see above comment). The limitations of the analysis are clearly described, and the authors provide a somewhat limited discussion of how these data can be be helpful to advance the impact of clinical trials for plague.

**Editorial and Data Presentation Modifications?**

Reviewer #1: (No Response)

Reviewer #2: Line 49: pestis should be written without upper-case letter

Line 225: range of temperature: 36°C to 103.2°C. Is it possible? or is it a problem of unit?

Table 1: References are missing for 5 studies:

Three cases of bubonic plague arising in England, 1916

Plague : Shasta County, California, 1965

Suspected Case of Imported bubonic plague, 1966

Bubonic Plague – California, 1970

Case report, Lazet, 2018

Reviewer #3: I have no additional editorial suggestions.

**Summary and General Comments**

Reviewer #1: Bourner J. et al. collected the published papers in English on bubonic plague and systematically review the clinical profile of this disease for understanding its clinical characterization. 87 reports published between 1902 and 2021 were finally included in this study. 1343 probable and confirmed cases of bubonic plague mainly reported in the United States, the United Kingdom, Vietnam, Zambia, and Israel, Libya, Madagascar, Mongolia, Tanzania, and Uganda. With these available data the authors summarized the baseline and post-baseline signs and symptoms, the patients’ outcome, and treatment. 

While providing valuable data for understanding the clinical feature of bubonic plague, this review also presents a limitation, which should be discussed in this manuscript. The data collection was only limited in papers in English. There are many reports on bubonic plague around the world in other languages, such as reports in Russian and Chinese. Much more patients have been missed from this statistical analysis. When discuss the limitation, Xu’s paper needs to be cited (Xu, et al. Nonlinear effect of climate on plague during the third pandemic in China. PNAS 2011 June 6.). 

The flowing review is also recommended to cite in the section of introduction. Yang et al. Zoonoses (2023) 3:5; DOI 10.15212/ZOONOSES-2022-0040 

It is recommended that the big Table 1 should be moved to supplements.

Reviewer #2: This systematic review is very interesting as it is based on bubonic plague cases reported during a long period, in different countries and continents. As few cases were described during the last years, this review gathers many informative descriptions of cases providing data on number, size, and location of bubos, on progression of symptoms and on treatment.

Identification of clinically relevant endpoints for plague at baseline is difficult as they may depend on the clinical status of the patient, the progression of the disease and the treatment. So, this review may help to determine some commonly identified endpoints which could be used in therapeutic trials.

Reviewer #3: This manuscript describes a meta analysis of available clinical information on plague patients, with the goal to compile a robust description of the clinical manifestation of disease and response to antibiotic treatment, in order to help inform endpoint selection for clinical trials on experimental drugs that are intended for use in combination with antibiotics. Overall this is an important goal and the results were interesting. This assessment adds to a number of recent similar papers that have surveyed patient data from plague. The impact of the work would be strengthened by graphical presentation of the data that are used to form the most pronounced conclusions that were discussed in the final section.

PLOS authors have the option to publish the peer review history of their article (what does this mean?). If published, this will include your full peer review and any attached files.

Reviewer #1: Yes: Ruifu Yang

Reviewer #2: No

Reviewer #3: No
---

## [Decision Letter · Decision Letter 1]

14 Oct 2023

Dear Ms Bourner,

We are pleased to inform you that your manuscript 'A systematic review of the clinical profile of patients with bubonic plague and the outcome measures used in research settings' has been provisionally accepted for publication in PLOS Neglected Tropical Diseases.

Best regards,

Vladimir L. Motin, Ph.D.

Guest Editor

Stuart Blacksell

Section Editor

The response to reviewers is satisfactory.

Reviewer's Responses to Questions

**Key Review Criteria Required for Acceptance?**

**Methods**

-Are the objectives of the study clearly articulated with a clear testable hypothesis stated?

-Is the study design appropriate to address the stated objectives?

-Is the population clearly described and appropriate for the hypothesis being tested?

-Is the sample size sufficient to ensure adequate power to address the hypothesis being tested?

-Were correct statistical analysis used to support conclusions?

-Are there concerns about ethical or regulatory requirements being met?

Reviewer #2: The objectives of the study are clear: the authors aim to identify endpoints that could be used as the primary efficacy endpoint in therapeutic trial on bubonic plague.

The study design seems appropriate. The criteria used to include studies are clear. The large number of included studies provides informative data on signs and symptoms.

Reviewer #3: (No Response)

**Results**

-Does the analysis presented match the analysis plan?

-Are the results clearly and completely presented?

-Are the figures (Tables, Images) of sufficient quality for clarity?

Reviewer #2: Presented analysis match the analysis plan.

Results are clearly and completely presented.

Figures and tables are clear.

Reviewer #3: (No Response)

**Conclusions**

-Are the conclusions supported by the data presented?

-Are the limitations of analysis clearly described?

-Do the authors discuss how these data can be helpful to advance our understanding of the topic under study?

-Is public health relevance addressed?

Reviewer #2: The conclusions are well supported by the data

the bias are presented in the results section

The discussion is interesting and may be helpful to advance in therapeutic trial for bubonic plague

Reviewer #3: (No Response)

**Editorial and Data Presentation Modifications?**

Reviewer #2: The corrections have been done

Reviewer #3: (No Response)

**Summary and General Comments**

Reviewer #2: This systematic review is very interesting as it is based on bubonic plague cases reported during a long period, in different countries and continents. As few cases were described during the last years, this review gathers many informative descriptions of cases providing data on number, size, and location of bubos, on progression of symptoms and on treatment.

Identification of clinically relevant endpoints for plague at baseline is difficult as they may depend on the clinical status of the patient, the progression of the disease and the treatment. So, this review may help to determine some commonly identified endpoints which could be used in therapeutic trials.

Reviewer #3: The authors present a revised manuscript that incorporates suggestions of the previous review. In addition, they have provided a rebuttal to some of the criticisms that did not result in revisions to the manuscript. In general, I don't agree with the points made in the rebuttal, however the criticisms were not aimed at the data directly, but rather the presentation of data to improve clarity. Therefore, since no substantive changes are needed, the revised manuscript is satisfactory in its attempt to address the criticisms of the original manuscript.

PLOS authors have the option to publish the peer review history of their article (what does this mean?). If published, this will include your full peer review and any attached files.

Reviewer #2: No

Reviewer #3: No

---

## [Editor Report · Acceptance letter]

26 Oct 2023

Dear Ms Bourner,

We are delighted to inform you that your manuscript, "A systematic review of the clinical profile of patients with bubonic plague and the outcome measures used in research settings," has been formally accepted for publication in PLOS Neglected Tropical Diseases.

Best regards,

Shaden Kamhawi

co-Editor-in-Chief

Paul Brindley

co-Editor-in-Chief
